# Capacity-Aware Mixture Law Enables Efficient LLM Data Optimization

**Jingwei Li** [* 1 2]  **Xinran Gu** [* 1 2]  **Jingzhao Zhang** [1 2]

## Abstract

A data mixture refers to how different data sources are combined to train large language models, and selecting an effective mixture is crucial for optimal downstream performance. Existing methods either conduct costly searches directly on the target model or rely on mixture scaling laws that fail to extrapolate well to large model sizes. We address these limitations by introducing a compute-efficient pipeline for data mixture scaling. First, we propose CAMEL, a capacity-aware mixture law that models validation loss with the nonlinear interplay between model size and mixture. We also introduce a loss-to-benchmark prediction law that estimates benchmark accuracy from validation loss, enabling end-to-end performance prediction for the target model. Next, we study how to allocate a fixed compute budget across model scales to fit the law and reduce prediction error. Finally, we apply our method to Mixture-of-Experts models with up to 7B-A150M parameters to fit the law, and verify the optimal mixture derived from the law by extrapolating to a 55B-A1.2B target model. Compared to prior methods, we reduce mixture optimization costs by 50% and improves downstream benchmark performance by up to 3%.

## 1. Introduction

Large language models (LLMs) are typically pretrained on mixtures of diverse domains, including general knowledge, code, math, and multilingual content (Doddapaneni et al., 2025; Li et al., 2023; Dubey et al., 2024; Team et al., 2023; Taylor et al., 2022). While the pretraining phase usually utilizes all available data without explicit tradeoffs, the scenario changes during *mid-training* phase (Sun et al., 2020;

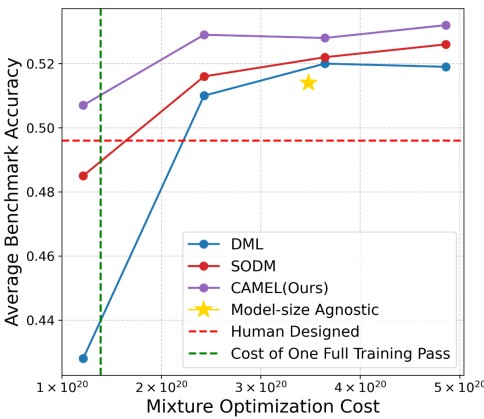

*Figure 1.* **Mixture optimization on the target model under different compute budgets.** We evaluate different mixture extrapolation methods by applying them to a larger target model with varying training FLOPS as mixture optimization costs. CAMEL, our proposed method, identifies high-quality data mixtures with even less than the cost of one full training pass on the target model. As the optimization budget increases, CAMEL achieves higher average benchmark accuracy than baseline methods while using less than 50% of the compute cost of the baseline.

Yıldız et al., 2024; Luo et al., 2025; Mendieta et al., 2023). Mid-training is a crucial period when the model quickly builds essential skills such as abstract reasoning, math problem solving, coding, and planning (Blakeney et al., 2024; Colombo et al., 2024; Chen et al., 2023; Hu et al., 2024). In cases like mid-training, data quality is prioritized over data quantity, as model ability can be easily weakened when mixed with low-quality, high-volume web text. Consequently, the allocation of domain-specific data becomes critical, and the mixture ratios significantly influence downstream performance. However, exhaustive exploration of these ratios is computationally expensive; hence, efficiently identifying optimal data mixtures is essential.

A growing body of work has explored data mixture optimization. Pioneering works search for optimal mixtures on small proxy models and then directly apply the resulting mixtures to larger models (Xie et al., 2023; Fan et al., 2024; Liu et al., 2024b). However, Magnusson et al. (2025); Gu et al. (2025); Mizrahi et al. (2025) show that mixtures optimized on small models do not necessarily transfer well to larger models, highlighting the need to explicitly account for model size in mixture design. To address this issue, another line of work

[*]Equal contribution  [1]Institute for Interdisciplinary Information Sciences, Tsinghua University  [2]Shanghai Qizhi Institute. Correspondence to: Jingwei Li <ljw22@mails.tsinghua.edu.cn>, Jingzhao Zhang <jingzhaoz@mail.tsinghua.edu.cn>.

*Proceedings of the 43[rd] International Conference on Machine Learning*, Seoul, South Korea. PMLR 306, 2026. Copyright 2026 by the author(s).

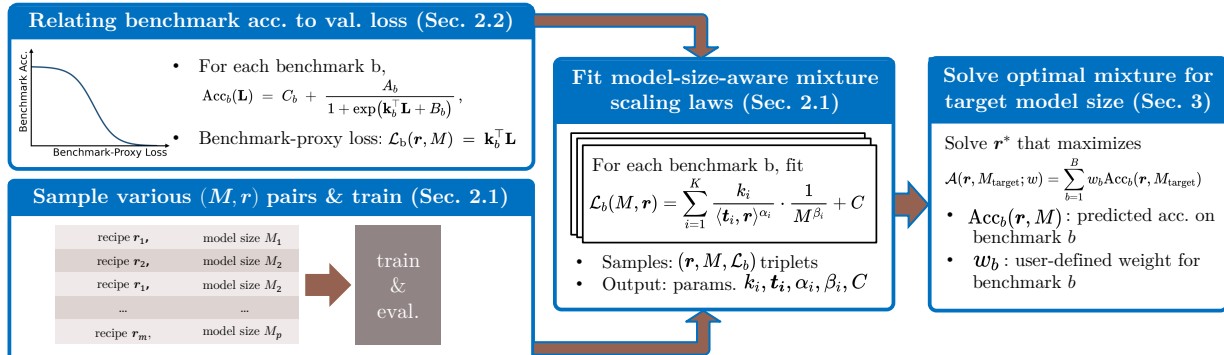

*Figure 2.* **End-to-end framework for data mixture extrapolation under model scaling.** We first fit a loss-to-benchmark mapping to relate validation loss to downstream benchmark accuracy (Section 2.2). We then model validation loss as a function of model size and data mixtures using sampled $(M, r)$ pairs from smaller models (Section 2.1). These components together enable extrapolation to large models and direct optimization of data mixtures for target-scale performance (Section 3).

adapts the scaling law framework to data mixture design by modeling the relationship between mixture and validation loss (Ye et al., 2025; Kang et al., 2024; Gu et al., 2024; Que et al., 2024; Shukor et al., 2025). However, few studies empirically validate the effectiveness of these predicted optimal mixtures when extrapolated to much larger models, such as those with up to 50B parameters.

Inspired by prior studies, this work aims to develop a practical procedure for optimizing data mixtures in LLM mid-training, in order to maximize downstream performance under minimal computation cost. We analyze the mixture optimization pipeline and propose targeted improvements.

First, we model data mixture optimization as a capacity allocation problem, following the perspective of (Gu et al., 2025). Solving this problem yields a data mixture scaling law that depends jointly on mixture ratios and model size. This contrasts with prior formulations that factor these effects into separable terms (Ye et al., 2025; Kang et al., 2024). The resulting law provides a more accurate description of mixture effects across model scales and reduces validation loss prediction error (Figure 5).

Second, while existing laws focus primarily on optimizing the validation loss (e.g. (Ye et al., 2025; Kang et al., 2024; Ge et al., 2024)), the loss does not always align with downstream performance (Lourie et al., 2025). Building on recent studies that link validation loss to benchmark accuracy (Dubey et al., 2024; Gadre et al., 2024; Magnusson et al., 2025; Chen et al., 2026), we introduce a loss-to-benchmark prediction law that maps multiple validation losses to each benchmark's downstream performance, achieving accurate benchmark prediction in practice (Figure 4).

Beyond scaling laws, we systematically investigate the sampling strategies under a limited compute budget. Unlike previous works that rely on fixed sampling heuristics, we analyze how to optimally allocate limited compute resources across different model sizes. Our findings indicate that allocation is critical; specifically, we derive an *hourglass* strategy with prioritizing samples at the smallest and largest model scales while reducing intermediate scales. As shown in Figure 7, this strategy consistently minimizes extrapolation error than other sampling strategies.

Finally, we apply our framework to train a larger model(55B-A1.2B). As shown in Figure 1, CAMEL identifies strong data mixtures using less than the cost of one full training pass on the target model, and achieves higher average benchmark accuracy than baseline methods at much lower cost.

Figure 2 illustrates the overall pipeline of our approach. In summary, our contributions are as follows:

- **Capacity-Aware Mixture Scaling Laws.** We derive a loss-prediction mixture law that jointly depends on mixture ratios and model size, enabling more accurate prediction than baseline methods. We further extend the law to directly predict benchmark accuracy.

- **Compute-Aware Experiment Design.** We propose an *hourglass* sample strategy for parameter estimation of the mixture scaling law. This strategy reduces prediction error under the same compute budget.

- **Verification by Extrapolating to Large Models.** We evaluate the optimal mixtures on larger models (up to 55B-A1.2B) and show that the mixtures derived by our law consistently achieve higher benchmark accuracy than baseline methods with less compute.

## 2. Data Mixture Scaling Laws

In this section, we introduce the design for our proposed data mixture scaling laws. We begin by formalizing the setup and introducing basic definitions. We consider training models with data from $n$ domains $\mathcal{D}_1, \ldots, \mathcal{D}_n$. A random sample $x$

is drawn from domain $\mathcal{D}_i$ with probability $r_i$, leading to the mixture distribution

$$p(x \mid \boldsymbol{r}) = \sum_{i=1}^{n} r_i \, p(x \mid \mathcal{D}_i),$$

where $\boldsymbol{r} = (r_1, \ldots, r_n)$ lies on the probability simplex $\Delta^{n-1} = \{r \in \mathbb{R}_{\geq 0}^n : \sum_i r_i = 1\}$.

Given a mixture $\boldsymbol{r}$, the model is trained by drawing samples from $p(\cdot \mid \boldsymbol{r})$ and minimizing the expected next-token prediction loss

$$\min_{\theta} \ \mathbb{E}_{x \sim p(\cdot \mid \boldsymbol{r})} \big[ - \log p_\theta(x) \big],$$

which produces parameters $\theta(\boldsymbol{r})$. We then evaluate these parameters on a held-out validation distribution $\mathcal{V}$, and denote the resulting loss as a function of $\boldsymbol{r}$:

$$\mathcal{L}_{\text{val}}(\boldsymbol{r}) = \mathbb{E}_{x \sim \mathcal{V}} \big[ - \log p_{\theta(\boldsymbol{r})}(x) \big].$$

Empirical scaling laws provide a general framework for modeling generalization performance, (Kaplan et al., 2020; Henighan et al., 2020; Hoffmann et al., 2022). Similarly, the scaling-law framework can be extended to data mixture design, enabling the prediction of optimal mixture ratios (Ye et al., 2025; Fan et al., 2024; Liu et al., 2024b; Shukor et al., 2025). Among these approaches, Ye et al. (2025) propose a representative Data Mixing Law (DML) that models validation loss as a function of mixture ratios. For a fixed model size and training step with $k$ intrinsic domains, the validation loss is formulated as:

$$\mathcal{L}_{\text{val}}(\boldsymbol{r}) = \sum_{i=1}^{k} S_i \left[ C_i + K_i \exp\left( \sum_{j=1}^{n} t_{ij} r_j \right) \right],$$

where $S_i, C_i, K_i, t_{ij}$ are learnable parameters. If this law can achieve a small estimation error on unseen mixtures, it provides a principled way to predict performance and optimize the data mixture. In particular, one can approximate the optimal mixture by solving

$$\boldsymbol{r}^* = \arg \min_{\boldsymbol{r} \in \Delta^{n-1}} \widehat{\mathcal{L}}_{\text{val}},$$

where $\widehat{\mathcal{L}}_{\text{val}}(\boldsymbol{r})$ is the fitted approximation of validation loss.

We note that DML models the relationship between data mixture and loss, but it requires an additional sample-complexity dimension to extrapolate to larger model sizes. Based on this observation, (Shukor et al., 2025) propose two model-size-aware scaling laws and select the better one according to empirical validation, which we refer to as SODM (Scaling Law for Optimal Data Mixture). Building on these works, we attempt to derive a scaling law from a capacity-aware view, as discussed below.

## 2.1. Capacity-Aware Mixture Scaling Laws

We develop a capacity-aware data mixture scaling law that captures the interaction between model size and data mixtures. Prior work shows that mixture effects depend on model scale (Gu et al., 2025; Mizrahi et al., 2025) so we explicitly incorporate model size into the scaling law. Compared with approaches that use model size and data mixture separately to fit the law (Ye et al., 2025; Kang et al., 2024), our method yields more accurate loss prediction (Figure 5).

**Intrinsic domains and mixture-induced domain weights.** Since validation data may come from real-world distributions with unknown compositions of explicit domains, we model them using intrinsic domains without requiring an explicit partition of the data. Intuitively, intrinsic domains can be viewed as a small set of latent components that explain variations in validation loss. For example, a knowledge dataset may be decomposed into components such as factual recall, reasoning, and language understanding. Following (Ye et al., 2025), we assume that both the training data and the validation data can be expressed through $k$ intrinsic domains. Each dataset $j$ is modeled as a distribution over these domains: let $t_{ij} \geq 0$ denote the proportion of domain $i$ in dataset $j$, satisfying $\sum_{i=1}^{k} t_{ij} = 1$ for all $j$. We define $\boldsymbol{t}_i = (t_{i1}, \ldots, t_{in})^\top \in \mathbb{R}^n$ as the domain-profile vector of intrinsic domain $i$ across datasets. Let $\boldsymbol{r} \in \Delta^{n-1}$ denote a mixture over $n$ datasets, where $r_j$ is the sampling weight of dataset $j$. The effective weight of intrinsic domain $i$ induced by mixture $\boldsymbol{r}$ is then given by

$$\eta_i(\boldsymbol{r}) := \langle \boldsymbol{t}_i, \boldsymbol{r} \rangle = \sum_{j=1}^{n} t_{ij} r_j.$$

**Domain-wise loss model.** We assume the training loss incurred on intrinsic domain $i$ follows a power law (Hoffmann et al., 2022). Let $\tilde{m}_i \geq 0$ denote the capacity allocated to domain $i$ (e.g., effective model parameters), we model the intrinsic-domain training loss as

$$\ell_i(\tilde{m}_i) := C_i + \frac{A_i}{\tilde{m}_i^{a_i}},$$

where $C_i, A_i, a_i$ are parameters which capture domain-specific learning difficulty.

**Capacity allocation objective.** Inspired by the capacity-allocation perspective of (Gu et al., 2025), we view pretraining as a process in which a model distributes its parameter capacity across intrinsic domains. We empirically examine this view by pretraining models of varying sizes on a fixed mixture of Math and Knowledge datasets, while explicitly monitoring the individual training loss for each domain.

As shown in Figure 3, increasing model size leads to a reduction in training loss for both datasets. For each domain

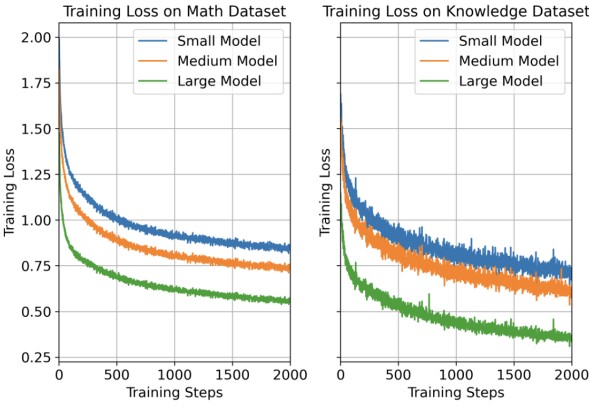

*Figure 3.* **Training loss observations for each domain across model sizes.** We train on a mixed dataset of math and knowledge and log the training loss for each domain. While larger models reduce loss in both areas, the rates of reduction differ significantly. This non-uniform scaling implies that the *effective parameters* allocated to each domain are redistributed dynamically rather than proportionally as the model scales.

$i$, we measure the loss improvements:

$$\Delta_i^{(M)} = \ell_i^{\text{small}} - \ell_i^{\text{medium}}, \qquad \Delta_i^{(L)} = \ell_i^{\text{small}} - \ell_i^{\text{large}},$$

and compute

$$R_i = \frac{\Delta_i^{(L)}}{\Delta_i^{(M)}}.$$

If effective model parameters scale proportionally as the model size scales, then we will have

$$\tilde{m}_i^{\text{medium}} = \lambda_M \tilde{m}_i^{\text{small}}, \qquad \tilde{m}_i^{\text{large}} = \lambda_L \tilde{m}_i^{\text{small}},$$

with shared scaling factors across domains. Under the scaling form $\ell_i(\tilde{m}_i) = C_i + \frac{A_i}{\tilde{m}_i^{\alpha_i}}$ with fixed training step, this implies

$$R_i = \frac{1 - \lambda_L^{-\alpha_i}}{1 - \lambda_M^{-\alpha_i}},$$

which should be similar across domains when $\alpha_i$ are close.

However, in Figure 3 the observed ratios differ significantly. At step 2000, we have

$$R_{\text{math}} \approx 2.4, \qquad R_{\text{know}} \approx 4.3.$$

Since $R_{\text{math}} \neq R_{\text{know}}$, this indicates that effective parameter allocation do not scale proportionally with model size. This disproportionate scaling behavior suggests that the *effective parameters* allocated to each intrinsic domain do not simply grow linearly with the total model size, but are instead *dynamically adjusted*.

These observations suggest an inherent capacity allocation strategy from pretraining. We hypothesize that given a total capacity budget $M$, the training process naturally finds an optimal resource distribution. Consequently, we model this implicit allocation as a constrained optimization problem, where the objective is to minimize the mixture-weighted sum of intrinsic losses by optimizing the effective parameters $\tilde{\boldsymbol{m}} = (\tilde{m}_1, \ldots, \tilde{m}_k)$ allocated to each domain:

$$\min_{\tilde{\boldsymbol{m}}} \ \mathcal{L}_{\text{train}}(\boldsymbol{r}; \tilde{\boldsymbol{m}}) := \sum_{i=1}^{k} \eta_i(\boldsymbol{r}) \, \ell_i(\tilde{m}_i)$$

$$= C_{\text{train}} + \sum_{i=1}^{k} \langle \boldsymbol{t}_i, \boldsymbol{r} \rangle \cdot \frac{A_i}{\tilde{m}_i^{a_i}}, \quad (1)$$

$$\text{s.t.} \ \sum_{i=1}^{k} \tilde{m}_i \leq M,$$

where $C_{\text{train}} = \sum_{i=1}^{k} \langle \boldsymbol{t}_i, \boldsymbol{r} \rangle C_i$ represents the irreducible component of training loss.

Optimizing this problem induces a capacity allocation that depends on the data mixture and the model size, which in turn determines the validation loss. Let $\tilde{\boldsymbol{m}}^* = (\tilde{m}_1^*, \ldots, \tilde{m}_k^*)$ denote the solution to the training optimization problem. The corresponding validation loss is given by

$$\mathcal{L}_{\text{val}}(\tilde{\boldsymbol{m}}^*) := C_{\text{val}} + \sum_{i=1}^{k} w_i \cdot \frac{A_i}{(\tilde{m}_i^*)^{a_i}}, \quad (2)$$

where $w_i$ is the weight of the $i$-th intrinsic domain in the validation distribution. To facilitate the analysis, we introduce the following approximation assumptions.

**Assumption 2.1.** We make the following assumptions about the parameters in (1).

1. **Nearly homogeneous learning-difficulty exponents.** There exists a common exponent $\bar{a} > 0$ and a perturbation vector $\boldsymbol{\varepsilon} = (\varepsilon_1, \ldots, \varepsilon_k)$ such that

$$a_i = \bar{a} + \varepsilon_i, \qquad \max_i |\varepsilon_i| \leq \epsilon_a,$$

where $\epsilon_a > 0$ is sufficiently small.

2. **Stable intrinsic-domain weights.** There exist reference weights $\bar{r}_i > 0$ and perturbations $\boldsymbol{\delta} = (\delta_1, \ldots, \delta_k)$ such that

$$\eta_i(\boldsymbol{r}) := \langle \boldsymbol{t}_i, \boldsymbol{r} \rangle = \bar{r}_i(1 + \delta_i), \qquad \max_i |\delta_i| \leq \epsilon_r,$$

where $\epsilon_r > 0$ is sufficiently small.

The first assumption is motivated by the empirical findings in (Hoffmann et al., 2022), which suggest that the learning-difficulty exponents across different domains are similar. Alternatively, considering that the exponent dictates how quickly the loss decreases, terms with significantly larger

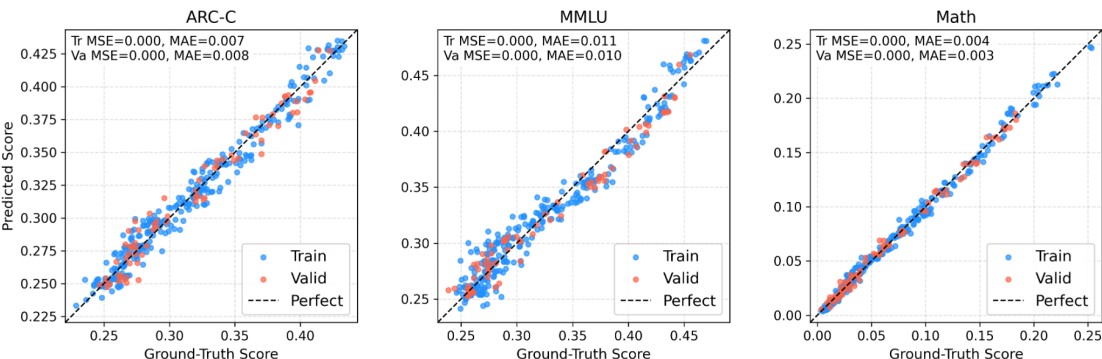

*Figure 4.* **Error of loss-to-benchmark prediction.** We model each downstream benchmark accuracy as a function of multiple validation losses. The scatter plots show predicted versus ground-truth scores on training and validation splits. The low prediction error demonstrates that validation losses can reliably predict downstream benchmark accuracy. See Appendix B.3 for details and results on other benchmarks.

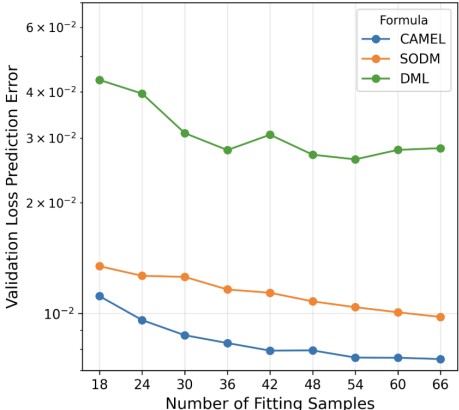

*Figure 5.* **Comparison between CAMEL and baseline scaling laws.** We compare the fitting error of our proposed *Capacity-Aware Mixture Law (CAMEL)* with two baseline methods, DML (Ye et al., 2025) and SODM (Shukor et al., 2025). CAMEL achieves consistently lower fitting error and exhibits more stable extrapolation behavior across model scales.

exponents vanish faster and can be disregarded. The second assumption follows the observations in (Ye et al., 2025) that the induced intrinsic-domain weights vary only mildly across different mixtures. Under these assumptions, we can establish the following result of our new law:

**Theorem 2.2.** *Assume Assumption 2.1 holds. Solving the optimal allocation $\tilde{m}^*$ for (1), the validation loss can be written as*

$$\mathcal{L}_{\text{val}}(\boldsymbol{r}, M) = C + \sum_{i=1}^{k} \frac{K_i}{\langle \boldsymbol{t}_i, \boldsymbol{r} \rangle^{\alpha_i} M^{\beta_i}},$$

*where $C$ captures higher-order terms under Assumption 2.1 and $\alpha_i$, $\beta_i$, and $K_i$ are functions of $\{(A_i, a_i, w_i)\}_{i=1}^{k}$ in (2).*

We refer to this formulation as the *Capacity-Aware Mixture Law* (CAMEL). This law unifies the effects of data mixture and model capacity within a single expression. The proof of Theorem 2.2 is provided in Appendix D.

To assess predictive accuracy, we evaluate the extrapolation performance of our law and baseline laws (Ye et al., 2025; Shukor et al., 2025). Figure 5 reports the evaluation setup and extrapolation error. Our method consistently outperforms the baselines across different training samples. Further details on the models and datasets are provided in Appendix B.1. We also conduct ablation studies on the number of intrinsic domains $k$ in Appendix B.2 and show that our algorithm is robust to the choice of $k$.

### 2.2. Modeling Downstream Performance

Existing scaling laws mainly predict validation loss, which is useful for pretraining but may not align with downstream benchmark accuracy. We therefore model benchmark accuracy directly as a function of validation losses.

Following Dubey et al. (2024); Bhagia et al. (2024), we adopt a logistic form and extend it to a multi-dimensional setting. For benchmark $b$, we define

$$\text{Acc}_b(\mathbf{L}) = C_b + \frac{A_b}{1 + \exp(\mathbf{k}_b^\top \mathbf{L} + B_b)}, \quad (3)$$

where $\mathbf{L} = (L_1, \ldots, L_N)^\top$ denotes validation losses from multiple datasets, and $A_b, B_b, C_b \in \mathbb{R}$, $\mathbf{k}_b \in \mathbb{R}^N$ are learnable parameters. This extension to multiple losses is motivated by the multifaceted nature of benchmarks (e.g., MMLU), which often demand a synthesis of diverse capabilities such as knowledge retention and reasoning.

We fit (3) on 14 popular benchmarks (see Figure 4 and Appendix B.3) and find that this law achieves low error on all benchmarks, indicating that validation losses provide sufficient signal for downstream accuracy prediction.

To combine this law with the model-size-aware mixture scaling law, we first fit the loss-to-benchmark mapping, then define a benchmark-proxy loss

$$\mathcal{L}_{\text{b}}(\boldsymbol{r}, M) = \mathbf{k}_b^\top \mathbf{L}.$$

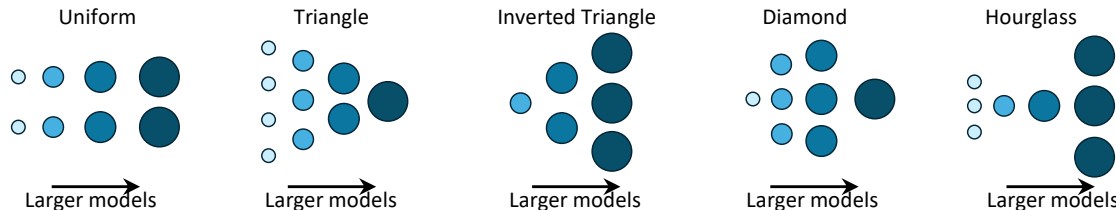

*Figure 6.* **Sampling strategies for fitting the scaling law.** We illustrate several strategies for selecting training configurations to fit the scaling law. Each subfigure corresponds to one sampling strategy, showing how combinations of model size and data mixture are chosen. Each circle represents one sampling point, and the circle area is proportional to the model size.

We use $\mathcal{L}_{\mathrm{b}}(\boldsymbol{r}, M)$, rather than individual losses, to fit the mixture-to-loss law. This choice reduces the number of parameters, since accurately fitting the loss-to-benchmark law requires over 200 validation sets, and directly incorporating all individual validation losses into the scaling law would introduce a large number of parameters and significantly increase optimization cost. Moreover, expressing benchmark performance in terms of a loss proxy preserves the homogeneity properties of prior loss-based scaling laws, allowing the two formulations to be combined consistently.

Together, the loss-prediction mixture law and the loss-to-benchmark law yield an end-to-end mapping

$$\boldsymbol{r} \;\mapsto\; \mathcal{L}_{\mathrm{b}}(\boldsymbol{r}, M) \;\mapsto\; \mathrm{Acc}_b(\boldsymbol{r}, M),$$

which directly predicts benchmark accuracy for a candidate mixture at model size $M$.

In practice, training may prioritize different objectives. We therefore define a weighted benchmark objective

$$\mathcal{A}(\boldsymbol{r}, M; w) \;=\; \sum_{b=1}^{B} w_b \, \mathrm{Acc}_b(\boldsymbol{r}, M),$$

with user-specified weights $w_b$. The optimal mixture is

$$\boldsymbol{r}^*(M; w) \;=\; \arg\max_{\boldsymbol{r} \in \Delta^{n-1}} \mathcal{A}(\boldsymbol{r}, M; w).$$

With the mixture-to-loss-to-benchmark scaling law, we fit parameters using data from small models and extrapolate to larger ones. Although this is cheaper than training large models, it still incurs nontrivial cost because each mixture requires a separate training run. This motivates compute-aware sampling across model scales under a fixed budget.

In practice, loss-to-benchmark data is cheap since a single training run gets multiple checkpoints and corresponding loss–accuracy pairs. In contrast, mixture-to-loss data is costly, so we focus on sampling strategies at this stage.

### 2.3. Sampling Strategies

Previous work (Ye et al., 2025; Ge et al., 2024; Shukor et al., 2025) typically adopts a *rectangle* allocation, drawing the

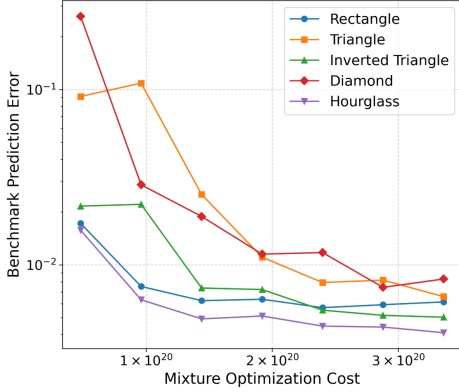

*Figure 7.* **Comparison of sampling strategies under fixed compute.** For each expression, five strategies are compared under varying mixture optimization cost. The **Hourglass** strategy consistently achieves the lowest prediction error.

same number of mixtures from each scale. However, under a fixed computing budget, this strategy may not be optimal. For example, allocating fewer points to larger models can free enough compute to sample more additional points at smaller scales. This leads us to a crucial question: *Which allocation strategy minimizes prediction error under limited compute?*

To address this, we compare five sampling strategies: *rectangle*, *triangle*, *inverted triangle*, *diamond*, and *hourglass*, as illustrated in Figure 6. They are summarized as follows:

- **Rectangle.** Allocate points evenly across all scales, cycling from smallest to largest.

- **Triangle.** Allocate more points to smaller models, and fewer to larger models.

- **Inverted Triangle.** Allocate more points to larger models and fewer to smaller models.

- **Diamond.** Allocate points starting from the middle scales and expand outward.

- **Hourglass.** Allocate points at both extremes first and then fill toward the center.

| Benchmark Scores \ Method | MMLU | ARC-C | BBH | GSM8K | MATH | HumanEval | C-Eval | Weighted Average Score |
|---|---|---|---|---|---|---|---|---|
| Human Designed | 0.573 | 0.535 | 0.429 | 0.652 | 0.325 | 0.354 | 0.637 | 0.496 |
| Model-size agnostic (Xie et al., 2023) | 0.569 | 0.529 | 0.437 | 0.674 | 0.336 | 0.450 | 0.627 | 0.514 |
| DML (Ye et al., 2025) | **0.579** | 0.552 | 0.454 | 0.696 | 0.393 | 0.353 | 0.631 | 0.519 |
| SODM (Shukor et al., 2025) | 0.566 | 0.541 | 0.453 | **0.705** | **0.399** | 0.421 | 0.628 | 0.526 |
| **Ours** | 0.577 | **0.553** | **0.455** | 0.693 | 0.372 | **0.451** | **0.653** | **0.532** |

*Table 1.* **Benchmark performance under the Balanced objective.** Our method achieves the highest weighted average score, computed using the benchmark optimization objective weights, and performs better than baseline mixtures on most benchmarks, indicating stable extrapolation to larger model scales.

We conduct experiments with different mixture optimization cost and sample strategies. As shown in Figure 7, the *Hourglass* strategy achieves the lowest prediction error when applied with our law, indicating that default sampling methods (*Rectangle*) are not optimal and more structured parameter estimation strategies can substantially reduce prediction error. Details of the experiments are in Appendix B.4.

## 3. Evaluation on Larger Models

We now evaluate our scaling pipeline on larger models. The procedure samples mixture–loss and loss–benchmark pairs using a compute-aware strategy, fits the scaling law, and extrapolates it to larger model sizes to obtain the optimal mixture. The experimental setup is described below.

### 3.1. Setup

**Model Architecture and Training.** We use the Deepseek V3 architecture (Liu et al., 2024a), instantiated at eight scales with a shared architecture. For fitting the mixture law, we use model sizes ranging from 590M-A12M to 7B-A150M. For evaluation, we use a larger model with 55B-A1.2B parameters. Full hyperparameter specifications and training details are provided in Appendix A.1 and A.2.

**Training Dataset and Mixtures.** Our training data includes five domains: English, Chinese, Code, Math, and Knowledge. We begin with a human-designed default mixture and create variants by upsampling or downsampling each domain. Details of dataset are given in Appendix A.3.

**Benchmarks.** We optimize data mixtures using seven benchmarks covering language understanding, reasoning, mathematics, and code generation: MMLU (Hendrycks et al., 2020), ARC-C (Clark et al., 2018), BIGBench-Hard (Suzgun et al., 2023), GSM8K (Cobbe et al., 2021), MATH (Hendrycks et al., 2021), HumanEval (Chen et al., 2021), and C-Eval (Huang et al., 2023). To evaluate generalization, we additionally report results on held-out benchmarks, including ARC-E (Clark et al., 2018), HellaSwag (Zellers et al., 2019), CMMLU (Li et al., 2024), TriviaQA (Joshi et al., 2017), CMATH (Wei et al., 2023), MBPP (Austin et al., 2021), and MGSM (Shi et al., 2023).

### 3.2. Experimental Design

We compare CAMEL with four baselines: (i) a **model-size-agnostic** method (Xie et al., 2023) that optimizes mixtures on small models and transfers them to larger ones; (ii) mixture predictions from prior scaling laws (Ye et al., 2025), denoted as **DML**; (iii) an alternative scaling-law formulation (Shukor et al., 2025), denoted as **SODM**; and (iv) a **human-designed** mixture based on the cooldown recipe, where we directly perform grid search over many recipes on the target model and select the best-performing recipe.

We fit end-to-end data mixture scaling laws under different training FLOPS as mixture optimization costs. CAMEL uses the *Hourglass* sampling strategy, while all baselines use the default *Rectangle* strategy. Details of the fitting datasets are provided in Appendix C.1.

We evaluate four training objectives: **Balanced**, **Math Specialized**, **Code Specialized**, and **Knowledge Specialized**. Each objective is defined as a weighted combination of benchmark accuracies, with weights given in Appendix C.2. Results are reported using the **Weighted Average Score** under the corresponding objective.

### 3.3. Results

**Performance on target benchmarks.** Figure 1 and Table 1 show that our extrapolated mixtures consistently outperform baseline methods across different compute budgets. Under the Balanced objective, our method achieves the highest weighted average score. These results demonstrate that CAMEL enables accurate, efficient mixture optimization.

**Robustness across diverse training objectives.** Beyond the Balanced objective, we evaluate CAMEL under Math, Code, and Knowledge Specialized targets. As shown in Table 2, CAMEL consistently achieves the highest weighted average score across all targets. This demonstrates that CAMEL reliably identifies optimal mixtures for both general-purpose and domain-specific training goals.

**Generalization to held-out tasks.** We evaluate generalization on held-out benchmarks not used during optimization (Figure 8). CAMEL achieves competitive performance

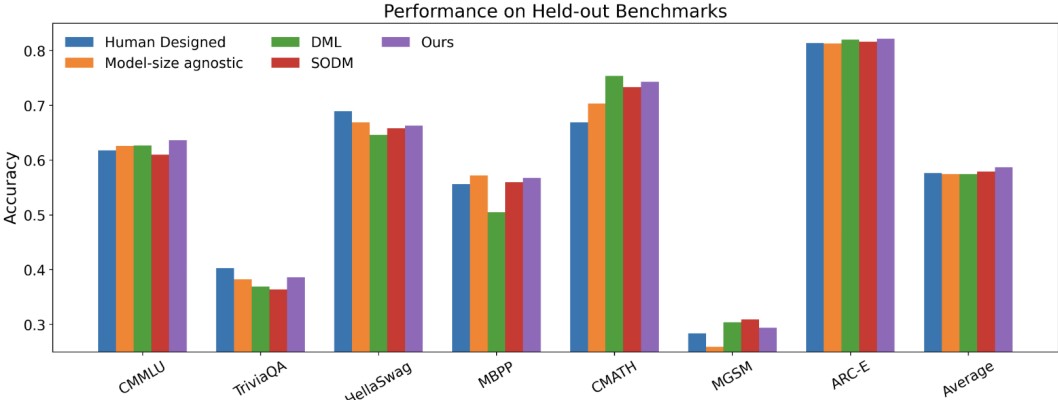

*Figure 8.* **Performance on held-out benchmarks.** Our method achieves the highest average accuracy on benchmarks not used during optimization, indicating strong generalization beyond the proxy objectives. This observation is consistent with prior work (Mizrahi et al., 2025) that mixtures optimized on diverse benchmarks generalize to unseen tasks.

|  | Specialized Target | Math | Code | Knowledge |
|---|---|---|---|---|
| Method | | | | |
| Model-size agnostic (Xie et al., 2023) | | 0.448 | 0.497 | 0.551 |
| DML (Ye et al., 2025) | | 0.479 | 0.500 | 0.550 |
| SODM (Shukor et al., 2025) | | 0.494 | 0.520 | 0.559 |
| **Ours** | | **0.497** | **0.528** | **0.565** |

*Table 2.* **Benchmark performance under other targets.** Our method achieves the highest weighted average score across all other specialized targets, outperforming the baselines and demonstrating robust mixture optimization under diverse training targets.

on all benchmarks and the highest average accuracy, indicating that it does not overfit the proxy objectives but instead learns mixtures that generalize across tasks.

**Evolution of the optimal mixture with model size.** We study how the optimal data mixture varies with model size. Figure 9 shows CAMEL-derived mixtures for a balanced target at different model sizes. As model size increases, the optimal weight on Knowledge increases, while those on Math and Code decrease. This suggests that larger models absorb general knowledge more efficiently, so knowledge data should be given more weight at larger scales. This finding provides valuable guidance for determining the optimal data mixture for different model sizes.

## 4. Related Work

**Data Mixture Optimization on Proxy Models.** A common approach is to optimize domain weights on small proxy models and transfer them to larger training. Xie et al. (2023) propose DoReMi, which reweights domains using group distributionally robust optimization (Group DRO) on a proxy model. Fan et al. (2024) refine this idea with DOGE, removing reliance on a reference model by directly estimating each domain's contribution, yielding mixtures that transfer more reliably. Liu et al. (2024b) adopt a predictive strategy, training regressors on small-model results to estimate

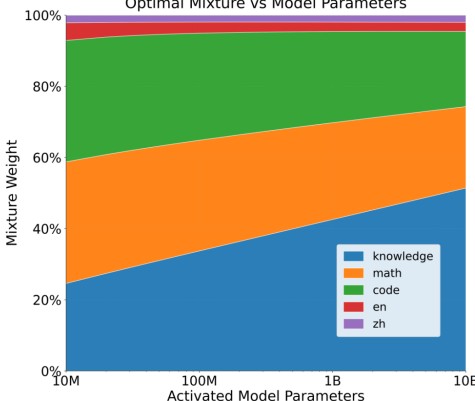

*Figure 9.* **Optimal mixture weights across model sizes.** The x-axis denotes activated model parameters and the y-axis shows dataset mixture weights. As model size increases, the proportion of Knowledge data grows, while Math and Code decrease, indicating stronger knowledge absorption in larger models.

unseen mixtures. Other variants cluster data into semantic groups (Diao et al., 2025) or vectorize datasets for mixture discovery without proxies (Zhang et al., 2025). However, Magnusson et al. (2025); Gu et al. (2025) show that mixtures optimized on small models may not scale well, underscoring the need to explicitly incorporate model size into mixture design.

**Loss-Prediction Mixture Laws.** Another line of work models how mixture ratios affect validation loss via parametric laws. Ye et al. (2025) and Kang et al. (2024) both decouple mixture interpolation from scale extrapolation: mixture effects are fitted on small models, while extrapolation to larger models is handled separately. Specifically, Ye et al. (2025) extrapolate to larger model sizes using a Chinchilla-style scaling law, whereas Kang et al. (2024) assume that optimal mixture ratios evolve exponentially with model size. Ge et al. (2024) propose a bivariate law capturing the joint effect of domain ratios and training steps.

For continual pretraining, Gu et al. (2024) and Que et al. (2024) study how to balance general and domain-specific data. These approaches provide systematic ways to extrapolate mixture effects, but such extrapolation is achieved by introducing an additional sample-complexity dimension that requires more data to support scaling, leading to higher data and compute costs. Most closely related to our work, Shukor et al. (2025) model data mixtures jointly with model size. We improve this design with a theory-motivated scaling law and systematically compare their performance in Section 3.

**Linking Validation Loss to Benchmark Accuracy.** Validation loss is often used as a proxy for downstream performance, though it does not always align with benchmark accuracy. This motivates end-to-end formulations that predict benchmark outcomes directly. Dubey et al. (2024); Bhagia et al. (2024) fit loss scaling laws with model size and data, then apply a sigmoidal mapping to predict benchmark accuracy, leveraging both final and intermediate checkpoints. Magnusson et al. (2025) adopt a similar two-step design with power-law fits and sigmoid mappings, exploring checkpoint selection strategies. Gadre et al. (2024) propose an exponential decay law directly linking loss to benchmark error, bridging perplexity-based scaling and benchmark prediction. While these methods incorporate benchmark accuracy, they do not consider data mixture design. Following Dubey et al. (2024), we adopt a logistic parametric form and extend it to the multi-dimensional setting.

## 5. Conclusion and Future Work

We presented CAMEL, a capacity-aware scaling law for data mixture optimization that links mixture ratios, model size, and benchmark accuracy under compute limits. CAMEL achieves lower prediction error than prior methods, and our analysis shows that structured fitting strategies such as the *hourglass* policy outperform uniform sampling under fixed budgets. In practice, CAMEL achieves efficient mixture optimization, saving at least 50% of the compute cost compared to previous baseline methods while delivering superior results. High-quality mixtures for the target model can often be found with even less than one full training pass.

Future work includes improving the formulation with alternative parametric forms or simple non-parametric models, aiming to achieve even more accurate extrapolation results. Additionally, we plan to study adaptive strategies that allocate mixture evaluations across model scales to improve efficiency under tight compute budgets.

## Impact Statement

This paper presents work aimed at advancing the field of machine learning by optimizing data mixtures for large language models (LLMs). By developing a more efficient and scalable method to determine the best mixture of domain-specific data, we reduce the computational cost required to achieve high benchmark performance. Our approach not only offers significant improvements in performance but also reduces the resources needed for model training, which could lead to better applications of LLMs, such as solving math problems, providing coding assistance, and knowledge retrieval.

## Acknowledgements

Jingzhao Zhang is supported by National Key R&D Program of China 2024YFA1015800 and Xiongan AI institute.

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

# A. General Implementation Details

In this section, we describe the general details of our setup, such as the model architecture, dataset and training.

## A.1. Details of Model Architectures

We construct a family of smaller models similar to Deepseek V3 architecture (Liu et al., 2024a). For a given model scale $M$, the corresponding architectural hyperparameters are set as follows:

- **Number of layers:** $M$.

- **Number of attention heads:** $M$.

- **Hidden size:** $112 \times M$.

- **FFN intermediate size:** $32 \times M$.

- **Number of experts:** 384.

- **Experts per token:** 8.

- **Shared experts:** 1.

- **Number of Dense layers:** 1.

- **Sparsity:** 48.

- **Expert Grouping:** No.

The total parameter count is computed using the *activated parameter counting function*, under the convention that `layers = heads`.

## A.2. Details of Training Hyperparameters

Training was performed using a cosine learning rate schedule with an initial learning rate of $8 \times 10^{-4}$ and a minimum learning rate of $8 \times 10^{-5}$. The learning rate was warmed up over the first $10^9$ tokens and then decayed over a total of $2 \times 10^{10}$ tokens. Models were trained with a global batch size of 320, using a micro-batch size of 2 per data-parallel worker.

Optimization used the Muon optimizer (Liu et al., 2025), a variant of AdamW, with $\beta_2 = 0.95$ and weight decay set to 0.1. Gradients were clipped to a maximum norm of 1.0. Training was conducted in bfloat16 precision, with selected operations such as gradient accumulation performed in fp32 for numerical stability. Dropout was disabled for both attention and hidden states.

## A.3. Details of Datasets

Our mid-training dataset consists of five broad domains, each targeting a distinct capability of large language model training:

- **English data.** Cleaned and deduplicated English web and narrative corpora, covering general knowledge, factual content, and diverse writing styles.

- **Chinese data.** Curated Chinese web and literary sources, including news articles and long-form texts, providing broad coverage of Chinese language usage.

- **Knowledge data.** Educational and reference-oriented corpora such as textbooks, academic materials, and structured knowledge resources, collected from publicly available sources to strengthen factual accuracy and conceptual understanding.

- **Code data.** High-quality programming corpora across multiple languages, including open-source repositories and synthetic code examples, designed to support code generation and program understanding.

- **Math data.** Mathematics-focused datasets consisting of problem–solution pairs, reasoning-heavy questions, and synthetic examples, with emphasis on mathematical reasoning and symbolic manipulation.

These domains are combined to form a unified mid-training corpus that jointly supports general language modeling, knowledge acquisition, programming ability, and mathematical reasoning.

For each model scale, training proceeds in two stages. We first pretrain the model up to its compute-optimal number of tokens, as estimated by scaling laws. This is followed by a cooldown stage with an additional 20B tokens, during which the data mixture is systematically varied.

Specifically, we begin from a human-designed reference mixture $\boldsymbol{r} = (r_1, r_2, r_3, r_4, r_5)$ over the five domains. To study the effect of mixture variations, we construct a set of perturbed mixtures by individually adjusting the proportion of each domain. For each domain $i$, we increase $r_i$ by 20% and renormalize the remaining domains proportionally so that the mixture sums to one. Similarly, we decrease $r_i$ by 20% and renormalize the remaining domains accordingly. Together with the reference mixture, this results in a total of 11 distinct data mixtures for each model scale.

These mixtures are used either to fit the proposed scaling law or to evaluate its predictive performance.

## B. Experimental Details of Section 2

This section provides additional details on the experimental setup used in Section 2.

### B.1. Prediction of Validation Loss

We consider a collection of models trained at different scales $M$, each with a fixed set of mixture ratios. Smaller models are used to fit the scaling laws, while a larger model is held out for evaluation. At each training scale, we evaluate multiple mixture ratios $r$, resulting in a set of $(r, M)$ pairs for both training and testing. The training set includes models with $M \in \{4, 5, 6, 7, 8, 10\}$, yielding $6 \times 11 = 66$ training points in total. The held-out test scale is $M = 12$.

To evaluate predictive performance, we vary the number of training points as $n \in \{18, 24, 30, 36, 42, 48, 54, 60, 66\}$ by uniformly sampling from the full training set. For each value of $n$, we repeat the sampling procedure 20 times and average the results to reduce variance. Extrapolation performance is then evaluated on the held-out test set.

### B.2. Ablation Study of Intrinsic Domains

We conduct an ablation study on the number of intrinsic domains. The results are shown in Figure 10.

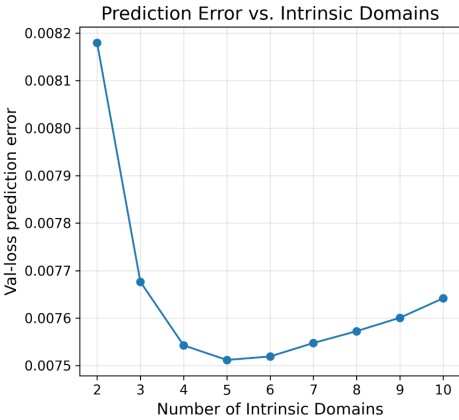

*Figure 10.* **Ablation study of intrinsic domains.** We vary the number of intrinsic domains $k$ and evaluate the validation-loss prediction error. When $k$ is small, the model has limited capacity, resulting in higher prediction error. As $k$ increases, the error decreases and reaches its minimum around $k = 5$. Further increasing $k$ leads to higher error, indicating overfitting and reduced robustness.

As shown in Figure 10, the optimal number of intrinsic domains is $k = 5$. When $k$ is too small, the decomposition cannot adequately capture the diversity of mixture effects. In contrast, an excessively large $k$ fragments the data across domains, which increases variance and degrades prediction accuracy due to overfitting. Therefore, we choose $k = 5$ in our law.

## B.3. Prediction of Benchmark Accuracy

We collect 100 checkpoints spanning multiple model scales and training steps. For each checkpoint, we record the validation losses on $N$ held-out validation sets together with the corresponding downstream benchmark accuracies. The validation losses are used as input features, while benchmark accuracy serves as the prediction target. The model is trained on 80 data points and evaluated on the remaining 20.

We further evaluate the formulation on a broad set of additional benchmarks covering language understanding, reasoning, mathematics, and code generation. Specifically, we consider MMLU (Hendrycks et al., 2020), ARC-C and ARC-E (Clark et al., 2018), BBH (Suzgun et al., 2023), GSM8K (Cobbe et al., 2021), MATH (Hendrycks et al., 2021), HumanEval (Chen et al., 2021), C-Eval (Huang et al., 2023), HellaSwag (Zellers et al., 2019), CMMLU (Li et al., 2024), TriviaQA (Joshi et al., 2017), CMATH (Wei et al., 2023), MBPP (Austin et al., 2021), and MGSM (Shi et al., 2023). Together, these benchmarks cover a wide range of skills, including factual recall, commonsense reasoning, professional knowledge, multi-step mathematical reasoning, and program synthesis.

As shown in Figure 11, the predicted accuracies produced by the fitted law closely match the observed results, resulting in consistently low prediction error. This indicates that the formulation generalizes well across diverse benchmarks and that the learned coefficients reliably capture the relationship between validation loss and downstream benchmark accuracy.

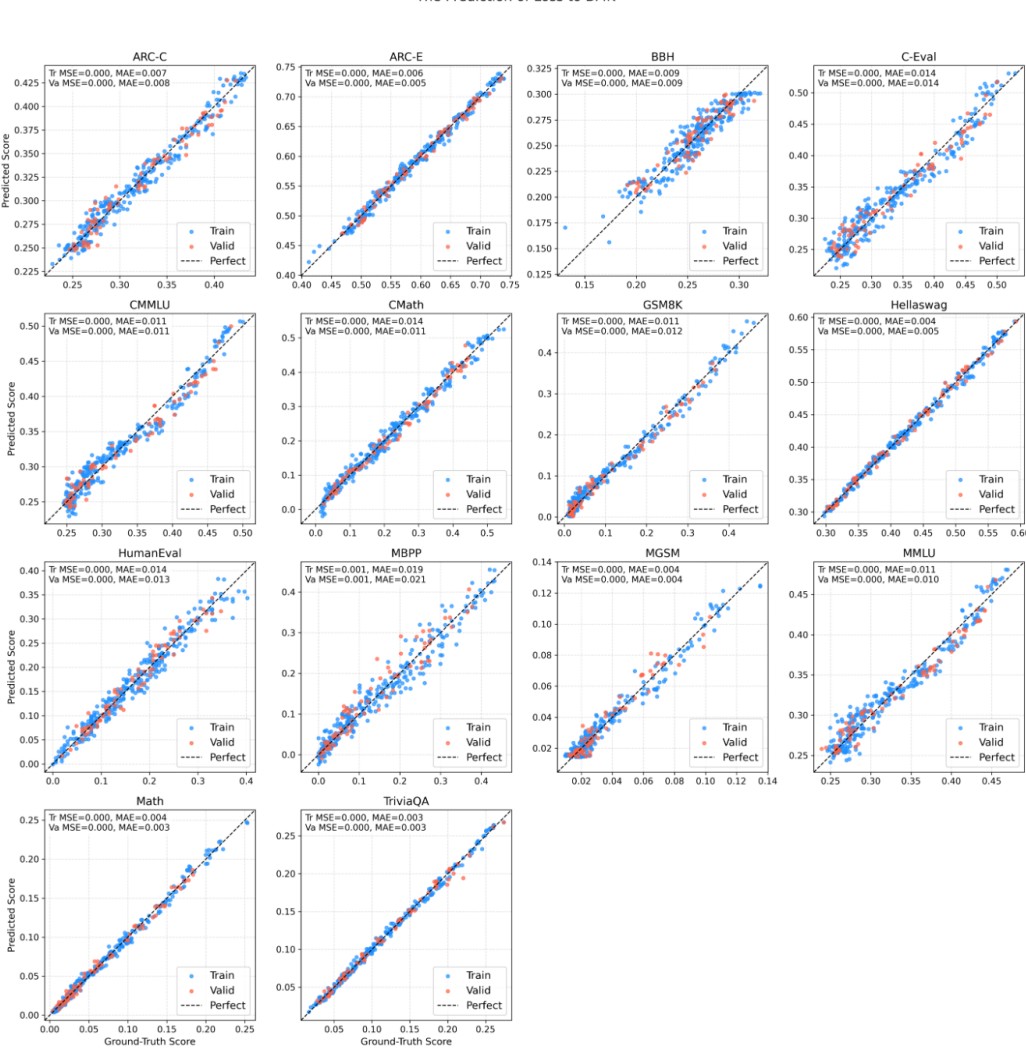

*Figure 11.* **Predictions of all benchmarks.** This figure shows predicted versus observed accuracies on all benchmarks. The close alignment indicates strong generalization of the proposed law.

## B.4. Experiments with Different Sampling Strategies

We use the same training and test datasets as described in Appendix B.1. For a given training budget, each sampling strategy first determines how many mixture configurations are selected at each model scale. Given this allocation, we then uniformly sample the specified number of mixtures from the 11 available mixture configurations at each model scale. This sampling process is repeated 10 times independently, and results are averaged to reduce sampling variance.

For each run, all sampled $(r, M)$ pairs across model scales are used to fit the scaling law. Extrapolation performance is evaluated by computing the prediction error on the held-out test scale. We report the average test error as well as the average number of training points per model scale over the 10 runs.

Training FLOPs are estimated as $6ND$, where $N$ denotes the number of model activated parameters and $D$ denotes the number of training tokens.

## C. Experimental Details of Section 3

This section provides additional details on the experimental setup used in Section 3.

### C.1. Details of Training Sets

We consider a set of models trained at different scales $M$, each with a fixed set of mixture ratios. At each training scale, we evaluate multiple mixture ratios $r$, yielding a collection of $(r, M)$ pairs used to fit the law. In particular, the scaling law is fitted using models with the same setup in Appendix B.1.

After fitting the scaling law, we extrapolate to the held-out scale $M = 20$. For each target configuration described in Section 3 (e.g., *Balanced*, *Math-specialized*), we construct a weighted objective by combining benchmark-specific predictions according to the corresponding target weights. We then substitute the target model scale $M = 20$ and the benchmark weights into the fitted law, and solve the resulting optimization problem to obtain the predicted optimal mixture ratios for that target configuration.

The predicted mixture is subsequently used to train a model at scale $M = 20$. We evaluate the trained model on downstream benchmarks and compare the resulting accuracies against those obtained by baseline mixture-selection methods.

### C.2. Details of Benchmark Weights

To construct the four training objectives described in Section 3.2, we assign different weights to evaluation benchmarks according to the target domain. The benchmarks considered include **MMLU** (Hendrycks et al., 2020), **ARC-Challenge** (Clark et al., 2018), **BBH** (Suzgun et al., 2023), **GSM8K** (Cobbe et al., 2021), **MATH** (Hendrycks et al., 2021), **HumanEval (pass@1)** (Chen et al., 2021), and **CEval** (Huang et al., 2023).

The *Balanced* configuration is designed to provide broad coverage across general knowledge, reasoning, mathematics, and code generation. We treat **CEval** and **ARC-Challenge** as proxies for Chinese and English factual and linguistic understanding, respectively, and assign them equal weights to maintain cross-lingual balance. **MMLU** and **BBH** capture broad academic knowledge and multi-task reasoning ability, and are therefore assigned slightly higher weights. The remaining benchmarks, namely **GSM8K**, **MATH**, and **HumanEval**, focus on numerical reasoning and program synthesis and receive the remaining weight mass. Table 3 summarizes the exact benchmark weights. This weighting scheme yields a stable, domain-agnostic training objective that avoids overemphasis on any single benchmark.

| Objective | MMLU | ARC-Challenge | BBH | GSM8K | Math | HumanEval (pass@1) | CEval |
|---|---|---|---|---|---|---|---|
| Balanced | 0.20 | 0.10 | 0.15 | 0.15 | 0.15 | 0.15 | 0.10 |
| Math Specialized | 0.12 | 0.08 | 0.12 | 0.15 | 0.30 | 0.15 | 0.08 |
| Code Specialized | 0.10 | 0.10 | 0.10 | 0.15 | 0.15 | 0.25 | 0.15 |
| Knowledge Specialized | 0.30 | 0.16 | 0.14 | 0.10 | 0.04 | 0.04 | 0.22 |

*Table 3.* Weights assigned to benchmarks under different training objectives. Each objective emphasizes a specific domain: the *Balanced* objective balances overall performance, the *Math Specialized* objective prioritizes mathematical reasoning, the *Code Specialized* objective emphasizes programming ability and the *Knowledge Specialized* objective emphasizes general knowledge ability.

## D. Proof of Theorem 2.2

In this section, we present the proof of our main theorem.

**Theorem 2.2.** *Assume Assumption 2.1 holds. Solving the optimal allocation $\tilde{\boldsymbol{m}}^*$ for (1), the validation loss can be written as*

$$\mathcal{L}_{\mathrm{val}}(\boldsymbol{r}, M) = C + \sum_{i=1}^{k} \frac{K_i}{\langle \boldsymbol{t}_i, \boldsymbol{r}\rangle^{\alpha_i} M^{\beta_i}},$$

*where $C$ captures higher-order terms under Assumption 2.1 and $\alpha_i$, $\beta_i$, and $K_i$ are functions of $\{(A_i, a_i, w_i)\}_{i=1}^{k}$ in (2).*

*Proof.* We prove the theorem in three steps.

**Step 1: KKT characterization.** Consider the training-loss optimization of (1). Its Lagrangian function is

$$\mathcal{L}(\boldsymbol{r}; \tilde{\boldsymbol{m}}; \lambda) = C_{\mathrm{train}} + \sum_{i=1}^{k} \frac{A_i \eta_i(\boldsymbol{r})}{\tilde{m}_i^{a_i}} + \lambda \Big(\sum_{i=1}^{k} \tilde{m}_i - M\Big), \qquad \lambda \geq 0.$$

Since each term $A_i \eta_i(\boldsymbol{r})/\tilde{m}_i^{a_i}$ is strictly decreasing in $\tilde{m}_i > 0$, any optimum must use the full budget, hence $\sum_{i=1}^{k} \tilde{m}_i = M$ and therefore by the complementary slackness condition we have $\lambda > 0$. Under the condition $\tilde{m}_i^{\star} > 0$ for all $i$, the KKT stationarity condition gives, for each $i$,

$$\frac{\partial \mathcal{L}}{\partial \tilde{m}_i} = -\frac{a_i A_i \eta_i(\boldsymbol{r})}{\tilde{m}_i^{a_i+1}} + \lambda = 0,$$

so we have

$$\tilde{m}_i^{\star} = \Big(\frac{a_i A_i \eta_i(\boldsymbol{r})}{\lambda}\Big)^{\frac{1}{a_i+1}}. \tag{4}$$

Plugging into the full budget constraint we get

$$\sum_{i=1}^{k} \Big(\frac{a_i A_i \eta_i(\boldsymbol{r})}{\lambda}\Big)^{\frac{1}{a_i+1}} = M. \tag{5}$$

**Step 2: Solving for $\lambda$ with Assumption 2.1.** Denote $p := \frac{1}{\bar{a}+1}$. By Assumption 2.1(1), for each $i$,

$$\frac{1}{a_i+1} = \frac{1}{\bar{a}+1+\varepsilon_i} = p + \Delta_i, \qquad |\Delta_i| \leq C\,\epsilon_a,$$

for a constant $C$ that depends only on $\bar{a}$.

Define

$$x_i := a_i A_i \eta_i(\boldsymbol{r}), \qquad L_x := \max_{1 \leq i \leq k} |\log x_i|.$$

Then we have

$$x_i^{p+\Delta_i} = x_i^p \exp(\Delta_i \log x_i) = x_i^p\Big(1 + O(\epsilon_a L_x)\Big),$$

uniformly over $i$. Similarly, defining $L_\lambda := |\log \lambda|$, we have

$$\lambda^{-(p+\Delta_i)} = \lambda^{-p} \exp(-\Delta_i \log \lambda) = \lambda^{-p}\Big(1 + O(\epsilon_a L_\lambda)\Big),$$

uniformly over $i$. Substituting these into (5) gives

$$M = \sum_{i=1}^{k} x_i^{p+\Delta_i} \lambda^{-(p+\Delta_i)} = \lambda^{-p} \sum_{i=1}^{k} x_i^p\Big(1 + O(\epsilon_a(L_x + L_\lambda))\Big). \tag{6}$$

Rearranging yields

$$\lambda^p = \frac{\sum_{i=1}^{k} x_i^p}{M} \Big( 1 + O(\epsilon_a (L_x + L_\lambda)) \Big). \tag{7}$$

By the definition of $\eta_i(\boldsymbol{r})$ and Assumption 2.1(2), we have

$$\eta_i(\boldsymbol{r}) = \bar{\boldsymbol{r}}_i (1 + \delta_i(\boldsymbol{r})), \qquad \max_i |\delta_i(\boldsymbol{r})| \le \epsilon_r.$$

Then for each $i$,

$$x_i^p = (a_i A_i \eta_i(\boldsymbol{r}))^p = (a_i A_i \bar{\boldsymbol{r}}_i)^p (1 + \delta_i(\boldsymbol{r}))^p = (a_i A_i \bar{\boldsymbol{r}}_i)^p \Big( 1 + O(\epsilon_r) \Big),$$

uniformly over $i$. Hence we get

$$\sum_{i=1}^{k} x_i^p = S_0 \Big( 1 + O(\epsilon_r) \Big), \qquad S_0 := \sum_{i=1}^{k} (a_i A_i \bar{\boldsymbol{r}}_i)^p. \tag{8}$$

Combining (7) and (8), we finally get

$$\lambda^p = \frac{S_0}{M} \Big( 1 + O(\epsilon_r) + O(\epsilon_a (L_x + L_\lambda)) \Big). \tag{9}$$

From (9), for sufficiently small $\epsilon_a, \epsilon_r$,

$$|\log \lambda| \le C_1 + \frac{1}{p} |\log(S_0/M)| =: L_0,$$

where $C_1$ depends only on $\bar{a}$. Plugging this back into (9), we have

$$\lambda^p = \frac{S_0}{M} \Big( 1 + O(\epsilon_r) + O(\epsilon_a (L_x + L_0)) \Big).$$

so we can finally get the solution of $\lambda$:

$$\lambda = \Big( \frac{S_0}{M} \Big)^{\bar{a}+1} \Big( 1 + O(\epsilon_r) + O(\epsilon_a (L_x + L_0)) \Big). \tag{10}$$

**Step 3: Representing validation loss as a function of $(\boldsymbol{r}, M)$.** From (4), we have

$$(\tilde{m}_i^\star)^{-a_i} = (a_i A_i)^{-\frac{a_i}{a_i+1}} \eta_i(\boldsymbol{r})^{-\frac{a_i}{a_i+1}} \lambda^{\frac{a_i}{a_i+1}}.$$

Define

$$\alpha_i := \frac{a_i}{a_i + 1}.$$

Then the $i$-th term in validation loss becomes

$$\frac{A_i w_i}{(\tilde{m}_i^\star)^{a_i}} = w_i A_i^{\frac{1}{a_i+1}} a_i^{-\frac{a_i}{a_i+1}} \cdot \eta_i(\boldsymbol{r})^{-\alpha_i} \cdot \lambda^{\alpha_i}. \tag{11}$$

Using (10), we get

$$\lambda^{\alpha_i} = \Big( \Big( \frac{S_0}{M} \Big)^{\bar{a}+1} \Big)^{\alpha_i} \Big( 1 + O(\epsilon_r) + O(\epsilon_a (L_x + L_0)) \Big).$$

Let $\Delta_i' := (\bar{a}+1)\alpha_i - a_i$, we have the following relation:

$$\Delta_i' = \frac{\bar{a} - a_i}{a_i + 1}, \qquad |\Delta_i'| \le C_2 \, \epsilon_a,$$

for a constant $C_2$ depending only on $\bar{a}$. Therefore,

$$\begin{aligned}
\Big( \frac{S_0}{M} \Big)^{(\bar{a}+1)\alpha_i} &= \Big( \frac{S_0}{M} \Big)^{a_i + \Delta_i'} = \Big( \frac{S_0}{M} \Big)^{a_i} \exp\big( \Delta_i' \log(S_0/M) \big) \\
&= S_0^{a_i} M^{-a_i} \Big( 1 + O(\epsilon_a |\log(S_0/M)|) \Big).
\end{aligned}$$

Combining the equilies, we yields

$$\lambda^{\alpha_i} = S_0^{a_i} M^{-a_i} \Big( 1 + O(\epsilon_r) + O(\epsilon_a(L_x + L_0)) \Big). \tag{12}$$

Plugging (12) into (11), we obtain

$$\frac{A_i w_i}{(\tilde{m}_i^\star)^{a_i}} = \frac{w_i A_i^{\frac{1}{a_i+1}} a_i^{-\frac{a_i}{a_i+1}} S_0^{a_i}}{\eta_i(\boldsymbol{r})^{\alpha_i} M^{a_i}} + O\big(\epsilon_r + \epsilon_a(L_x + L_0)\big).$$

Summing over $i = 1, \ldots, k$, we obtain the final representation of the validation loss:

$$\mathcal{L}_{\text{val}}(\boldsymbol{r}, M) = C + \sum_{i=1}^{k} \frac{K_i}{\eta_i(\boldsymbol{r})^{\alpha_i} M^{\beta_i}} = C + \sum_{i=1}^{k} \frac{K_i}{\langle \boldsymbol{t}_i, \boldsymbol{r} \rangle^{\alpha_i} M^{\beta_i}}$$

where

$$\alpha_i = \frac{a_i}{a_i + 1}, \qquad \beta_i = a_i, \qquad K_i = w_i A_i^{\frac{1}{a_i+1}} a_i^{-\frac{a_i}{a_i+1}} S_0^{a_i}, \qquad C = C_{\text{val}} + O\big(\epsilon_r + \epsilon_a(L_x + L_0)\big).$$

$\square$

