# OpenReview forum: "Capacity-Aware Mixture Law Enables Efficient LLM Data Optimization"
_ICML.cc/2026/Conference — ICML 2026 regular_

### Official Review · Reviewer_4WGk · 2026-03-10

**Soundness:** 3
**Presentation:** 3
**Significance:** 3
**Originality:** 3
**Overall Recommendation:** 4
**Confidence:** 3

**Summary:**

The paper presents CAMEL, a capacity-aware mixture law for LLM mid-training. The key idea is to model validation loss jointly as a function of model size and data mixture from a capacity-allocation perspective, rather than handling scaling and mixture design separately. The paper further introduces a loss-to-benchmark prediction law that maps validation losses to downstream benchmark accuracy, enabling direct optimization of data mixtures for end-to-end target performance. In addition, it studies compute-aware experiment design under a fixed budget and proposes an hourglass sampling strategy for selecting small-scale training runs used to fit the law. Experimentally, the method is fitted on smaller MoE models trained across multiple scales and mixture variants over several domains. The paper compares CAMEL against prior mixture-transfer and scaling-law baselines, and evaluates the extrapolated mixtures on a substantially larger target model under balanced and specialized objectives, as well as on held-out benchmarks.

**Compliance With Llm Reviewing Policy:**

Affirmed.

**Key Questions For Authors:**

Please refer to the above weak points W1-W4.

**Limitations:**

Yes

**Strengths And Weaknesses:**

Pros:

S1: The paper is well-written with clear motivation and well-structured pipeline. The proposed components (mixture law, benchmark mapping, and sampling strategy) are presented in a coherent progressive framework, and all theorems are proved.

S2: The problem is strongly motivated and highly relevant to the current LLM training paradigm. Efficient data-mixture optimization for mid-training is an important practical bottleneck, and the paper directly targets reducing expensive target-scale mixture search.

S3: The experimental study verifies both effectiveness and efficiency of CAMEL across different settings。

Cons:

W1: The theoretical justification of CAMEL relies on approximation assumptions that may be strong in practice, such as nearly homogeneous domain exponents and stable intrinsic-domain weights. Discussing how sensitive the final mixture recommendations are when these assumptions are violated would help quantify the scaling law more clearly.

W2: The empirical validation is conducted only on the DeepSeek family. It would strengthen the paper to evaluate CAMEL on other open-source models (e.g., Qwen), as well as on other architectures, different domain taxonomies, and richer mixture search spaces.

W3: The comparison between CAMEL and the baselines is somewhat confounded by the use of different sampling strategies: CAMEL adopts the hourglass strategy, whereas the baselines use the default rectangle strategy. A cleaner comparison would evaluate all methods under the same sampling budget and sampling strategy, in order to isolate gains from the scaling law itself from gains due to experimental design.

W4: Could some implementation details be made publicly available for reference?

---

> ### Author Rebuttal · Authors · 2026-03-30
>
> We sincerely thank the reviewer for the comments and suggestions. As suggested, we add additional experiments on new setup and different sampling strategies. We address the questions below.
>
> >Q1: The theoretical justification of CAMEL relies on approximation assumptions that may be strong in practice.
>
> **A1:** We thank the reviewer for this comment. The assumption of homogeneous learning-difficulty exponents is motivated by empirical findings in [1], which show similar exponents across domains, while the stable intrinsic-domain weights assumption follows observations in [2] that these weights vary only mildly across mixtures. These assumptions are mainly introduced to facilitate theoretical analysis and are not required to strictly hold in practice. Empirically, the derived scaling law performs consistently well across all experimental setups, indicating that the effectiveness of CAMEL remains robust even when these assumptions are not exactly satisfied.
>
> >Q2: It would strengthen the paper to evaluate CAMEL on other setups.
>
> **A2:** To evaluate CAMEL beyond the mid-training setting, we include additional pre-training experiments where all methods are trained from scratch and evaluated under both model size and training step extrapolation.
>
> We extend our scaling law to incorporate training steps:
>
> $$
> L_{\text{val}}(r, M, T) = C + \sum_{i=1}^{k} \frac{K_i}{\langle t_i, r \rangle^{\alpha_i} M^{\beta_i}} + K_T \cdot T^{-\gamma},
> $$
>
> where $T$ denotes the number of training steps, $K_T, \gamma$ are learnable parameters.
>
> We conduct experiments on the Nemotron CC dataset, which consists of four domains (Nemotron-SFT, Nemotron-Code, Nemotron-CC-Math, Nemotron-CC) and 13 subsets in total. We randomly sample 100M tokens from each subset to construct 13 i.i.d. validation sets. The remaining data are used for training.
>
> Starting from the reference mixture provided by Nemotron CC, we construct perturbed mixtures by reducing each domain by 10% and proportionally redistributing the removed mass to other domains. Together with the original reference mixture, this results in 5 mixture settings. We fit the scaling laws using MoE models with activated parameters of 3M, 11M, 22M, and training steps of 500, 1000, 2000, yielding 45 training points. We then extrapolate to a larger model (41M) and longer steps (4000-6000).
>
> We evaluate performance by comparing the predicted and actual validation losses across the 13 validation sets. Results show that our method achieves lower average prediction error than [1][2] when extrapolating across both model size and training steps:
>
> |Step|4000|5000|6000|
> |:---:|:----:|:----:|:----:|
> |DML[1]|0.132|0.160|0.187|
> |SODM[2]|0.089|0.092|0.101|
> |**Ours**|**0.084**|**0.089**|**0.093**|
>
> >Q3: The comparison between CAMEL and the baselines is somewhat confounded by the use of different sampling strategies.
>
> **A3:** We thank the reviewer for this comment. We conduct additional ablations where all methods are evaluated under the same sampling strategy. Specifically, we use the same experimental setup in Section 2.3, fit and evaluate the benchmark prediction error for both CAMEL and the baselines under (i) the hourglass strategy and (ii) the rectangle strategy.
>
> We first report the benchmark prediction error results under the hourglass strategy:
>
> |Mixture Optimization Cost|0.8e20|1.6e20|2.4e20|
> |:---:|:----:|:----:|:----:|
> |DML[1]|0.0191|0.0113|0.0097|
> |SODM[2]|0.0124|0.0076|0.0063|
> |**Ours**|**0.0064**|**0.0052**|**0.0044**|
>
> We then report results under the rectangle strategy:
>
> |Mixture Optimization Cost|0.8e20|1.6e20|2.4e20|
> |:---:|:----:|:----:|:----:|
> |DML[1]|0.0153|0.0107|0.0089|
> |SODM[2]|0.0127|0.0088|0.0065|
> |**Ours**|**0.0077**|**0.0068**|**0.0047**|
>
> The results show that the performance gains of CAMEL are not due to the choice of sampling strategy, but are intrinsic to the scaling law itself.
>
> >Q4: Could some implementation details be made publicly available for reference?
>
> **A4:** We appreciate the reviewer’s suggestion. We will provide more detailed implementation descriptions in the next version of the paper to improve clarity and reproducibility.
>
> We thank the reviewer once again for the valuable and helpful suggestions. We hope our responses have resolved the questions. If so, we kindly invite the reviewer to consider raising the rating. If there are any further questions or clarifications needed, we would be happy to provide additional details.
>
> **References**
>
> [1] Hoffmann, Jordan, et al. "Training compute-optimal large language models." Proceedings of the 36th International Conference on Neural Information Processing Systems. 2022.
>
> [2] Ye, Jiasheng, et al. "Data Mixing Laws: Optimizing Data Mixtures by Predicting Language Modeling Performance." The Thirteenth International Conference on Learning Representations.
>
> [3] Shukor, Mustafa, et al. "Scaling Laws for Optimal Data Mixtures." The Thirty-ninth Annual Conference on Neural Information Processing Systems.

---

> > ### Author Rebuttal · Reviewer_4WGk · 2026-04-04
> >
> > My concerns are all resolved.

---

> > > ### Author Response · Authors · 2026-04-04
> > >
> > > Thank you for your feedback and for taking the time to review our rebuttal! We will incorporate all the additional clarifications and experiments discussed in the rebuttal into the next version of the paper. We kindly ask you to reconsider the score based on the updated understanding. If you have any further questions or concerns, we would be happy to address them during the remaining discussion period.

---

### Official Review · Reviewer_UCjd · 2026-03-13

**Soundness:** 3
**Presentation:** 3
**Significance:** 3
**Originality:** 3
**Overall Recommendation:** 5
**Confidence:** 5

**Summary:**

The paper introduces CAMEL, a scaling law framework designed to jointly model the interactions between data mixture ratios and model capacity. Unlike prior works that often treat these variables in isolation, CAMEL characterizes their combined impact on validation loss and employs a separate logistic mapping to project these losses onto downstream benchmark accuracies. A key contribution of the paper is the exploration of compute-aware sampling, where the authors propose an *hourglass* allocation strategy, prioritizing the extremes of the model-scale spectrum under a fixed budget to improve fitting reliability. The framework is empirically validated on Mixture-of-Experts (MoE) architectures ranging from 7B to 55B parameters, demonstrating superior extrapolation accuracy and a 50% reduction in search costs compared to baseline laws like DML and SODM.

**Compliance With Llm Reviewing Policy:**

Affirmed.

**Final Justification:**

The response clearly addresses my concerns regarding the potential risk of overfitting CAMEL to a specific benchmark set, as well as how robustly the loss-to-benchmark mapping generalizes across different scales. I find the experimental analysis thorough in this paper, particularly the discussion on computation cost and the comparison of different sampling strategies for fitting the law. The proposed method is also practical for real-world use. I will maintain a positive score and increase my confidence by one point. I recommend acceptance.

**Key Questions For Authors:**

1. Could you report (or summarize in an appendix) the exact compute budgets for each method? Specifically, it would be great to see the FLOPs/tokens per sample, the number of samples per scale, the total search compute, or the final training compute.

2. How robustly does the loss-to-benchmark mapping generalize across different scales? A brief, intuitive discussion on whether mappings calibrated on smaller scales hold perfectly at larger scales would be very interesting.

3. Could you clarify the exact definition of M used in the scaling equations and disambiguate the notation for the final version?

**Limitations:**

The authors discuss the limitations of their work primarily within the "Conclusion and Future Work" section.

**Strengths And Weaknesses:**

Strengths:

A major strength of this work is its direct applicability to a critical bottleneck in LLM development: the computationally expensive search for optimal mid-training data mixtures. The authors propose CAMEL, a well-motivated capacity-aware scaling law that demonstrably improves prediction accuracy over existing mixture laws. I particularly appreciate the coherence of the end-to-end pipeline, specifically the elegant integration of a multi-loss-to-benchmark mapping with compute-aware sampling across various model scales.

Rather than relying on heuristics, CAMEL serves as a highly practical and principled tool for the community. It significantly reduces the compute-intensity of mid-training mixture selection while providing actionable, theoretical guidance on how optimal data distributions shift as model capacity increases. Furthermore, the empirical gains demonstrated on the large target model are consistent, structurally sound, and firmly establish the real-world utility of this approach. Overall, this is a strong contribution that I expect will be widely adopted.

Weaknesses:

One potential consideration with the loss-to-benchmark mapping is the theoretical risk of inadvertently overfitting the mixture design to a specific benchmark set. If the intrinsic domains and validation corpora are not fully representative, deploying such mixtures might subtly reduce downstream diversity or fairness.

There are a few minor instances where notation could be slightly streamlined for the reader's convenience. For example, the symbol M is occasionally used to denote both the number of layers/heads in the architecture description and the activated parameter size elsewhere. The context always makes the intended meaning clear, but a quick disambiguation in the camera-ready version would further polish this already well-written manuscript.

---

> ### Author Rebuttal · Authors · 2026-03-30
>
> We thank the reviewer for the positive feedback and constructive suggestions! In the following, we address the main concern raised.
>
> >Q1: One potential consideration with the loss-to-benchmark mapping is the theoretical risk of inadvertently overfitting the mixture design to a specific benchmark set.
>
> **A1:** We appreciate the reviewer’s question. The choice of benchmark set is user-driven, as different applications may prioritize different capabilities (e.g., balanced performance vs. math/code-focused performance). Our framework allows flexible specification of benchmark weights according to these needs.
>
> In our experiments, we use a diverse set of benchmarks as the optimization target to reduce the risk of overfitting to any single domain. We also evaluate the resulting mixtures on many benchmarks that are not included in the optimization objective (see Figure 8). The mixtures selected by our method also achieve strong performance on these unseen benchmarks and the average benchmark scores outperform those obtained by baseline methods. This suggests that our approach does not overfit to the chosen benchmark set and generalizes well beyond it. This observation is also consistent with prior work[1], which shows that mixtures optimized on diverse benchmarks tend to generalize to unseen tasks.
>
> >Q2: Could you report the exact compute budgets for each method?
>
> **A2:** We appreciate the reviewer’s question. We estimate the compute for each sample as $6ND$, where $N$ is the number of activated parameters of the training model for that sample, and $D$ is the number of training tokens.
>
> For sampling across scales, the number of samples per scale is determined by the target mixture optimization cost and the sampling strategy. Specifically, for each scale, we select $m$ samples according to the allocated budget and randomly draw them from a pool of 11 mixture candidates (see Appendix B.4 for details).
>
> The reported training compute is the sum of the compute over all sampled training runs. We do not include additional costs such as search or law fitting, as these are relatively small compared to the training cost.
>
> In Figure 1 and the experiments in Section 3, we control the total training compute at $1.2e20$, $2.4e20$, $3.6e20$, and $4.8e20$. We will provide a more detailed explanation of the compute including per-scale allocation and total compute budgets in the next version of our paper.
>
> >Q3: How robustly does the loss-to-benchmark mapping generalize across different scales?
>
> **A3:** We thank the reviewer for raising this important question. To evaluate the robustness of the loss-to-benchmark mapping across model scales, we conduct an additional experiment following the setup in Section 2.2.
>
> We partition the (loss, benchmark accuracy) pairs by model scale into two groups: small-scale models (M = 4, 5, 6, 7, 8, 9; 244 pairs) and large-scale models (M = 10, 12, 14; 116 pairs). We fit the mapping using only the small-scale data and evaluate it on the large-scale data. The validation error comparisons are as follows:
>
> |Method|ARC-Challenge|Helleswag|BBH|
> |:---:|:----:|:----:|:----:|
> |Random Split across All Scales|0.008|0.005|0.009|
> |Train on Small and Test on Large Scales|0.009|0.007|0.009|
>
> We find that the validation errors under the cross-scale split are comparable to those under random partitioning across all scales. This suggests that the loss-to-benchmark mapping maintains similar accuracy across model scales. We will include this discussion in the next version of the paper.
>
> >Q4: The symbol M is occasionally used to denote both the number of layers/heads in the architecture description and the activated parameter size elsewhere. Could you clarify the exact definition of M used in the scaling equations and disambiguate the notation for the final version?
>
> **A4:** We thank the reviewer for pointing out this ambiguity. In all scaling equations, $M$ denotes the number of activated parameters of the model. We will revise the notation in Appendix A.1 to make this definition explicit and consistently use $M$ for activated parameter size, while using separate symbols (e.g., $L$ for number of layers) in the architecture description to avoid confusion.
>
> We thank the reviewer once again for the valuable and helpful suggestions. We would be happy to provide further clarifications if the reviewer has any additional questions.
>
> **References**
>
> [1] Mizrahi, David, et al. "Language models improve when pretraining data matches target tasks." arXiv preprint arXiv:2507.12466 (2025).

---

> > ### Author Rebuttal · Reviewer_UCjd · 2026-04-04
> >
> > Thank you for the detailed response. The response clearly addresses my concerns regarding the potential risk of overfitting CAMEL to a specific benchmark set, as well as how robustly the loss-to-benchmark mapping generalizes across different scales. I find the experimental analysis thorough in this paper, particularly the discussion on computation cost and the comparison of different sampling strategies for fitting the law. The proposed method is also practical for real-world use. I will maintain a positive score and increase my confidence by one point.

---

> > > ### Author Response · Authors · 2026-04-04
> > >
> > > We thank the reviewer for the positive feedback. We will incorporate the relevant experiments and additional discussions into the next revision of the paper.

---

### Official Review · Reviewer_HnKR · 2026-03-13

**Soundness:** 2
**Presentation:** 2
**Significance:** 3
**Originality:** 2
**Overall Recommendation:** 4
**Confidence:** 5

**Summary:**

The paper proposes CAMEL, a mixing law that captures both the data mixture ratios and the size of the model. Many previous approaches have focused on learning a mix at a small size and then directly transferring that mix to a larger model. However, the value of different domains changes across model sizes, so it is important to model this. The paper derives a bi-variate mixing law that takes as input $r$ (ratios) and $M$ (model size), and outputs predicted performance on a downstream benchmark. Then, a (weighted) average predicted benchmark score is optimized to get a final (mid-training) data mix. The paper also finds that an "hourglass" sampling strategy across model sizes helps learn a good mixture law, and verifies downstream performance on models of size 55B parameters.

**Compliance With Llm Reviewing Policy:**

Affirmed.

**Final Justification:**

The rebuttal addressed my concerns by providing more baselines and more exploration of the behavior of $m^\star$.

**Key Questions For Authors:**

1. Can you provide further validation of m*? I.e., that training implicitly solves the optimal allocation problem and that m* is not all uniform.
2. How does the proposed method, along with DML, SODM, etc. perform for pre-training? Reporting on smaller models, or simply measuring regression fit, would be sufficient. This would be a fairer setting of comparison against existing mixing laws.
3. Can you compare against methods that do not account for model size and directly transfer mixture ratios? (maybe in the pre-training setting, since most existing methods are designed for the pre-training setting).

**Limitations:**

Yes.

**Strengths And Weaknesses:**

**Strengths**

Soundness:
- The paper provides motivation on why model size should be modeled, and how the proposed mixture law has a good regression fit. These empirical findings provide additional support beyond downstream performance of the proposed data mixture.

Significance:
- The paper evaluates data mixing at large model sizes (55B), which has not been done before to my knowledge.

Originality:
- The paper makes a novel contribution in identifying how one should sample from models of different sizes to learn the bi-variate mixing law. This is a scientifically interesting and practically useful result around an understudied lever of mixing/scaling laws.
- While previous parametric mixing law papers have mostly focused on optimizing domain validation loss, this paper extends optimization to downstream benchmarks by fitting a sigmoid function.

**Weaknesses**

Soundness:
- I am curious about modeling the number of training steps too. In practice, larger models are pre/mid-trained on many more tokens, so altogether there would be three things to model: ratio, model size, and number of tokens/steps.
- It would be good to see more validation around the proposed model of implicit parameter allocation, m*. From your regression fits, what is the empirically estimated m*? Does this align with how domains are learned differently as models scale?
- Approaches like DML and SODM are focused on pre-training and do well in those settings. I think it is a bit misleading to compare directly to them, especially because you only mention that you are doing mid-training in the introduction and in the appendix. It would also be interesting to see how the mixture law changes as you start mid-training at different checkpoints, since it isn't always clear when one should start mid-training.
- I would encourage comparing against other approaches that do not account for model size such as RegMix (or CLIMB, which iteratively applies it in the mid-training setting), or Olmix (which uses a modification of the DML mixing law and fits directly on benchmarks without the val loss to benchmark accuracy conversion).

Presentation:
- I suggest more explicitly saying that this paper is focused on mid-training data mixing. DML, SODM, etc. are all for pre-training.
- The writing around section 2 could be signposted better. 2.1 starts with a lot of motivating assumptions (the hypothesis of implicit parameter allocation), and finally the equation in theorem 2.2 is the actual law that is being fit.
- I am not sure how to interpret Figure 3. Is the point being made around how knowledge vs math have different slopes of training loss? Different gaps across small/medium/large? Different absolute loss?

Originality:
- There exist works (like DML, Autoscale) that already account for both the data mixture ratios and the model size. There also exist works that fit to downstream benchmarks, like CLIMB and Olmix. However, I do think that the hourglass result is novel, and the hypothesis that different domains have different implicit parameter allocation is novel and interesting.

---

> ### Author Rebuttal · Authors · 2026-03-30
>
> We thank the reviewer for the detailed and constructive feedback. We include additional experiments:
> 1. We extend our study to the pre-training setting, including extrapolation over training steps.
> 2. We propose an empirical method to illustrate that $m_i^*$ does not scale proportionally with model size.
> 3. We include comparisons with a model-agnostic baseline, RegMix.
>
> Below, we address the main questions.
>
> >Q1: How to model the number of training steps?
>
> **A1:** We thank the reviewer for this question. In our setting, we fix the number of training steps, as we focus on the mid-training stage where the token budget is relatively small compared to model size.
>
> To extrapolate over training steps, we consider an  extension by adding an extra power-law term:
> $$
> L_{\text{val}}(r, M, T) = C + \sum_{i=1}^{k} \frac{K_i}{\langle t_i, r \rangle^{\alpha_i} M^{\beta_i}} + K_T \cdot T^{-\gamma},
> $$
> where $T$ denotes the number of training steps, and $K_T, \gamma$ are learnable parameters.
>
> Empirically, we find that this simple additive form already provides a good fit in the pre-training setting (see A2 of Reviewer 4WGk for details).
> A more detailed study of training steps is left for future work.
>
> >Q2: It would be good to see more validation around the proposed model of implicit parameter allocation m*.
>
> **A2:** We thank the reviewer for this question. We provide an estimation method to show that $m_i^*$ across domains does not scale proportionally with model size.
>
> We use the setup in Figure 3 with the training step fixed at 2000. For each domain $i$, we measure the loss improvements:
> $$
> \Delta_i^{(M)} = L_i^{\text{small}} - L_i^{\text{medium}}, \qquad
> \Delta_i^{(L)} = L_i^{\text{small}} - L_i^{\text{large}},
> $$
> and compute
> $$
> R_i = \frac{\Delta_i^{(L)}}{\Delta_i^{(M)}}.
> $$
>
> If effective parameters scale proportionally, then
> $$
> m_i^{\text{medium}} = \lambda_M m_i^{\text{small}}, \qquad
> m_i^{\text{large}} = \lambda_L m_i^{\text{small}},
> $$
> with shared scaling factors across domains. Under the scaling form $L_i(m_i) = C_i + \frac{A_i}{m_i^{\alpha_i}}$ with fixed training step, this implies
> $$
> R_i =
> \frac{1-\lambda_L^{-\alpha_i}}{1-\lambda_M^{-\alpha_i}},
> $$
> which should be similar across domains when $\alpha_i$ are close.
>
> However, in Figure 3 the observed ratios differ significantly. At step 2000, we have
> $$
> R_{\text{math}} \approx 2.4, \qquad
> R_{\text{know}} \approx 4.3.
> $$
> Since $R_{\text{math}} \neq R_{\text{know}}$, this indicates that effective parameter allocation does not scale proportionally with model size. We will add this analysis to the main text.
>
> >Q3: How does the method perform for pre-training?
>
> **A3:** We thank the reviewer for the question. We conduct additional experiments in the pre-training setting and observe consistent improvements over baselines. Due to space constraints, we refer to A2 of Reviewer 4WGk for detailed results.
>
> >Q4: How does the mixture law change as you start mid-training at different checkpoints?
>
> **A4:** We thank the reviewer for this question. As pre-training progresses, different capabilities tend to saturate, and the resulting mid-training mixture law becomes more stable.
>
> In practice, models are typically trained well beyond the compute-optimal regime, and rarely remain undertrained. Our setting, which starts from a Chinchilla-optimal checkpoint (training tokens ≈ 20× parameters), serves as a reasonable choice.
>
> Studying different mid-training starting points requires running multiple full pre-training checkpoints, which is computationally expensive. We leave this direction for future work.
>
> >Q5: I would encourage comparing against other approaches that do not account for model size such as RegMix.
>
> **A5:** We thank the reviewer for the suggestion. We add a comparison with RegMix under the same setup as in A3. We implement RegMix[1] using a LightGBM model trained on all 22M-scale samples, with the objective defined as the average of the 13 validation losses. The optimized mixture is then evaluated by training a 41M model and measuring the average validation loss at step 4000.
>
> Our method achieves lower average validation loss than RegMix:
>
> |Method|Average Validation Loss|
> |:---:|:----:|
> |Regmix[1]|2.03|
> |**Ours**|**1.94**|
>
> >Q6: I am not sure how to interpret Figure 3.
>
> **A6:** We thank the reviewer for the question. Figure 3 illustrates that as model size increases, the validation loss reduction of fixed training step differs significantly across domains. A more detailed quantitative analysis is provided in A2. We will incorporate this explanation into the paper to make Figure 3 clearer.
>
> Finally, we thank the reviewer for the valuable suggestions. We hope our responses address the concerns and kindly invite the reviewer to consider raising the rating. We are happy to discuss further if needed.
>
> **Reference**
>
> [1] Liu, Qian, et al. "RegMix: Data Mixture as Regression for Language Model Pre-training." The Thirteenth International Conference on Learning Representations.

---

> > ### Author Rebuttal · Reviewer_HnKR · 2026-04-06
> >
> > Thank you for the response. It has addressed my concerns by providing additional baselines like RegMix and more exploration of $m^\star$. I will increase my score.

---

> > > ### Author Response · Authors · 2026-04-06
> > >
> > > Thank you for your feedback and for carefully reviewing our rebuttal! We will incorporate the relevant experiments and additional discussions into the next revision of the paper.

---

### Official Review · Reviewer_Ct7D · 2026-03-16

**Soundness:** 3
**Presentation:** 3
**Significance:** 3
**Originality:** 3
**Overall Recommendation:** 4
**Confidence:** 3

**Summary:**

The authors propose and empirically validate CAMEL, a model capacity-aware data mixture law. CAMEL models validation loss as a function of both data mixture and model size. Additionally, the authors propose to predict downstream benchmark performance from validation losses. Empirically, the authors show that CAMEL outperforms two other baselines for validation loss prediction, and predicted scores for select downstream tasks are highly correlated with actual ground truth downstream benchmark scores

**Compliance With Llm Reviewing Policy:**

Affirmed.

**Final Justification:**

My primary concerns were about presentation, which the rebuttal mostly addressed. Increasing my presentation score (2 -> 3). I appreciate the improved contextualization around claims of increased "efficiency," but I hesitate to increase my score or confidence due to the substantial computational costs of fitting a mixture law and the limited comparison to alternative dynamic approaches that do not rely on fitting a general mixture or scaling law (see first and third related works). My overall assessment remains positive

Additional relevant citations:
https://arxiv.org/abs/2410.11820, https://arxiv.org/abs/2410.15661, https://arxiv.org/abs/2411.05735, https://dl.acm.org/doi/abs/10.1145/3786668

**Key Questions For Authors:**

1. What implicit assumptions are at play when the authors describe "intrinsic domains"? Can there be overlap? How large is a domain? Is there some notion of a reasonable range of $k$?
2. Can the authors elaborate on the efficiency of their proposed methods?

**Limitations:**

yes

**Strengths And Weaknesses:**

Strengths:
1. CAMEL is empirically strong and can accurately predict downstream performance directly given a candidate data mixture rather than just validation losses
2. Potentially very high impact problem to tackle with

Weaknesses:
1. Certain assumptions and definitions are under-specified. See questions below
2. I would have liked to see more discussion of compute costs of running a predictive -- especially given that efficiency is a key enough contribution that it is mentioned in the conclusion that CAMEL saves at least 50% of the compute cost of previous methods. See questions below

Minor or inline comments:
1. line 034: "we reduces" -> "we reduce"
2. 074 (right column): "to train larger model" -> to train a larger model?
3. line 202: "capacity allocation depends" -> "capacity allocation that depends"?

---

> ### Author Rebuttal · Authors · 2026-03-30
>
> We sincerely appreciate the reviewer's feedback and suggestions. We will correct minor typos in the revised manuscript. Below, we provide responses to the main questions.
>
> >Q1: What implicit assumptions are at play when the authors describe "intrinsic domains"? Can there be overlap? How large is a domain? Is there some notion of a reasonable range of $k$?
>
> **A1:** We thank the reviewer for the question. The concept of intrinsic domains was introduced in [1], and we adopt it as a latent modeling tool in our formulation. Since validation data may come from real-world distributions with unknown compositions of explicit domains, we model them using intrinsic domains without requiring an explicit partition of the data.
>
> Intuitively, intrinsic domains can be viewed as a small set of latent components that explain variations in validation loss. For example, a knowledge dataset may be decomposed into components such as factual recall, reasoning, and language understanding. These components are not tied to disjoint subsets of tokens; instead, a single data point can contribute to multiple components, so overlap is naturally allowed.
>
> Formally, this assumes that the validation loss can be approximated as a sum of a few such components, where each component follows a similar scaling behavior with respect to data mixture and model size. The “size” of an intrinsic domain is therefore not defined by a fixed subset of data, but is reflected by its mixture weight.
>
> Regarding the choice of $k$, we conduct an ablation study in Figure 10 (Appendix B.2) and find that the prediction performance is not sensitive to its exact value. In our setup, the best performance is achieved at $k=5$, which matches the mixture dimension $n$. This suggests that setting $k$ on the order of $n$ is sufficient to obtain stable and accurate predictions. We will add this discussion to the paper.
>
>
> >Q2: Can the authors elaborate on the efficiency of their proposed methods?
>
> **A2:** We thank the reviewer for this question. We demonstrate the efficiency of our method in Figure 1 and describe the computation of the mixture optimization cost in Appendix B.4.
>
> As shown in Figure 1, under a mixture optimization cost of $2.4e20$, our method achieves an average benchmark score of $0.529$. In comparison, under a mixture optimization cost of $4.8e20$, DML [1] and SODM [2] achieve average benchmark scores of $0.519$ and $0.526$, respectively. This shows that our method achieves better performance while reducing mixture optimization compute by at least 50%.
>
> We will include a more detailed discussion of training efficiency in the next version of the paper.
>
> We thank the reviewer again for the helpful feedback. We hope our responses have resolved the questions. If so, we kindly invite the reviewer to consider raising the rating. Please feel free to let us know if there are any further questions.
>
> **References**
>
> [1] Ye, Jiasheng, et al. "Data Mixing Laws: Optimizing Data Mixtures by Predicting Language Modeling Performance." The Thirteenth International Conference on Learning Representations.
>
> [2] Shukor, Mustafa, et al. "Scaling Laws for Optimal Data Mixtures." The Thirty-ninth Annual Conference on Neural Information Processing Systems.

---

> > ### Author Rebuttal · Reviewer_Ct7D · 2026-04-04
> >
> > I appreciate the engagement with my questions on domain definition. Ye et al do not describe "intrinsic domains" explicitly, nor does the present study as submitted seem to present a clear working definition of the term; but including a description along the lines of the elaboration provided in the rebuttal will help the presentation
> >
> > The intuition of "roughly $k=n$" is helpful. I realize now that my original question had to do with $n$, the number of datasets that a mixture is found over, vs the parameter $k$ that is ablated in Figure 10 -- **this is my own oversight -- the authors adequately answered my question as asked,** and I do not expect that this would have been empirically tested, but I am still curious to understand whether there is any intuition on (a rough order of magnitude of) what the limit of reasonable $n$ could be
> >
> > On efficiency, I would like to sincerely apologize for my underspecified comments. My original concerns are unaddressed, largely due to my own failure to specify. Concretely:
> >
> > 1. Figure 1 lacks units. In fact, ctrl+f for "FLOPs" (the presumed unit used as a catch-all stand-in for "cost") shows only a single result in the Appendix
> > 2. Is there any overhead associated with use of the CAMEL framework? Even if not training FLOPs -- e.g. repeatedly computing multiple validation losses and downstream task performance/loss during training can incur nontrivial costs, and I would like to understand whether this is something relevant to CAMEL
> > 3. Contextualizing FLOPs with notions of cost more directly relevant to stakeholders (e.g. GPU hours, wall clock time, and/or $, data samples (as Reviewer UCjd mentioned)) can improve presentation greatly
> >
> > I will hold my score as is for now

---

> > > ### Author Response · Authors · 2026-04-04
> > >
> > > We thank the reviewer for the response. Regarding the scale of $n$, its practical upper bound is mainly determined by computational budget. A larger $n$ increases the dimensionality of the mixture space, requiring more sampled mixtures to reliably fit the scaling law, and thus higher cost. At the same time, a larger $n$ corresponds to a finer-grained data partition, which can in principle yield better mixtures.
> > >
> > > In our setup, we use $n=5$ coarse categories (math, code, knowledge, English, Chinese), each containing multiple sub-datasets with fixed internal proportions. The optimized mixture over categories can then be mapped to sub-datasets. For example, a data source such as Wikipedia would correspond to one sub-dataset within the “knowledge” category. Using sub-datasets directly (larger $n$) is possible and may improve performance, but would require substantially more computation.
> > >
> > > Regarding Figure 1, the reported cost corresponds to training FLOPs. We thank the reviewer for pointing this out and will clarify this explicitly in the next revision.
> > >
> > > For the loss-to-benchmark mapping, there is some additional overhead from evaluating checkpoints on downstream benchmarks. However, as noted at the end of Page 5, this cost is much smaller than training new models, since a single training run provides multiple checkpoints and corresponding (loss, benchmark accuracy) pairs. Moreover, these evaluation results can be accumulated and reused in practice, further reducing the effective marginal cost.
> > >
> > > In our setup, all (loss, benchmark accuracy) pairs are obtained by evaluating checkpoints from existing training runs in our experiments under different mixtures. Therefore, no additional training is required, and the only extra cost comes from running benchmark evaluations. The total evaluation data across all benchmarks is about 10M tokens; using the standard $6ND$ FLOPs estimate, this corresponds to approximately $2.9 \times 10^{16}$ FLOPs, which is negligible compared to the $10^{20}$ scale reported in Figure 1. We thus omit this cost in our analysis.
> > >
> > > From a wall-clock perspective, collecting all training samples in our experiments takes about 7 days on 64 A100 GPUs. In practice, the required compute can be substantially lower, depending on the target model scale and the desired level of improvement. As shown in Figure 1, using only one-quarter of the total training FLOPs (approximately 42 hours on 64 A100 GPUs) already yields a strong mixture with competitive performance gains, indicating that strong solutions can be obtained with significantly reduced cost.
> > >
> > > We thank the reviewer again for the helpful suggestions. We will incorporate these discussions in the next revision of our paper.

---

### Decision · Program_Chairs · 2026-04-30

**Decision:**

Accept (regular)

**Comment:**

This paper proposes CAMEL, a capacity-aware mixture law that jointly models validation loss as a function of data mixture ratios and model size, addressing the limitation of existing methods that fail to extrapolate effectively to larger scales. The authors further introduce a mapping from validation loss to downstream benchmark accuracy and an "hourglass" sampling strategy to optimize compute allocation during the law-fitting phase. The method is validated on MoE models up to 55B parameters, demonstrating reduced optimization costs and improved downstream performance compared to baselines like DML and SODM.

The reviewers generally agree on the practical significance and novelty of modeling the interplay between model capacity and data mixture, particularly for mid-training scenarios. Reviewer UCjd (Score: 5) highlighted the coherent end-to-end pipeline and strong empirical gains. Reviewers Ct7D, HnKR, and 4WGk (Score: 4, 4, 4 respectively) raised concerns regarding the clarity of definitions (e.g., "intrinsic domains"), the specificity of compute cost reporting, comparisons with pre-training focused baselines, and potential confounding factors in sampling strategies.

During the rebuttal, the authors provided clarifications on the latent nature of intrinsic domains, detailed compute breakdowns (FLOPs and wall-clock time), and conducted additional experiments. They extended evaluations to the pre-training setting, added comparisons with RegMix under consistent sampling strategies, and demonstrated the robustness of the loss-to-benchmark mapping across scales. These responses addressed the reviewers' concerns, leading all three reviewers to confirm their issues were resolved and maintain or improve their positive assessments.

While some limitations remain regarding the theoretical assumptions and the scope of architectures tested (primarily DeepSeek/MoE), the empirical evidence supports the method's effectiveness. The paper offers a technically sound and practically useful tool for efficient data mixture optimization in large-scale LLM training.